# Accuracy and Precision of the Receptorial Responsiveness Method (RRM) in the Quantification of A_1_ Adenosine Receptor Agonists

**DOI:** 10.3390/ijms20246264

**Published:** 2019-12-12

**Authors:** Adrienn Monika Szabo, Gabor Viczjan, Tamas Erdei, Ildiko Simon, Rita Kiss, Andras Jozsef Szentmiklosi, Bela Juhasz, Csaba Papp, Judit Zsuga, Akos Pinter, Zoltan Szilvassy, Rudolf Gesztelyi

**Affiliations:** 1Department of Internal Medicine, Faculty of Medicine, University of Debrecen, H-4032 Debrecen, Hungary; szabo.adrienn23@gmail.com; 2Doctoral School of Nutrition and Food Sciences, University of Debrecen, H-4032 Debrecen, Hungary; 3Department of Pharmacology and Pharmacotherapy, Faculty of Medicine, University of Debrecen, H-4032 Debrecen, Hungary; vicgabaa@gmail.com (G.V.); erdei.tamas@pharm.unideb.hu (T.E.); simonildiko87@gmail.com (I.S.); kiss.rita@med.unideb.hu (R.K.); ajszm948@gmail.com (A.J.S.); juhasz.bela@med.unideb.hu (B.J.); szilvassy.zoltan@med.unideb.hu (Z.S.); 4Department of Health Systems Management and Quality Management for Health Care, Faculty of Public Health, University of Debrecen, H-4032 Debrecen, Hungary; dr.papp.csaba@gmail.com (C.P.); zsuga.judit@med.unideb.hu (J.Z.); 5Institute of Mathematics, Faculty of Science and Technology, University of Debrecen, H-4032 Debrecen, Hungary; apinter@science.unideb.hu

**Keywords:** A_1_ adenosine receptor, atrium, heart, nonlinear regression, receptorial responsiveness method, RRM

## Abstract

The receptorial responsiveness method (RRM) is a procedure that is based on a simple nonlinear regression while using a model with two variables (X, Y) and (at least) one parameter to be determined (c_x_). The model of RRM describes the co-action of two agonists that consume the same response capacity (due to the use of the same postreceptorial signaling in a biological system). While using RRM, uniquely, an acute increase in the concentration of an agonist (near the receptors) can be quantified (as c_x_), via evaluating E/c curves that were constructed with the same or another agonist in the same system. As this measurement is sensitive to the implementation of the curve fitting, the goal of the present study was to test RRM by combining different ways and setting options, namely: individual vs. global fitting, ordinary vs. robust fitting, and three weighting options (no weighting vs. weighting by 1/Y^2^ vs. weighting by 1/SD^2^). During the testing, RRM was used to estimate the known concentrations of stable synthetic A_1_ adenosine receptor agonists in isolated, paced guinea pig left atria. The estimates were then compared to the known agonist concentrations (to assess the accuracy of RRM); furthermore, the 95% confidence limits of the best-fit values were also considered (to evaluate the precision of RRM). It was found that, although the global fitting offered the most convenient way to perform RRM, the best estimates were provided by the individual fitting without any weighting, almost irrespective of the fact whether ordinary or robust fitting was chosen.

## 1. Introduction

Receptor theory is one of the most prominent concepts in pharmacology [1], with the great advantage of underlying a variety of methods suitable for quantitative analysis [2]. A way to determine constants (or sometimes variables) characterizing receptors, ligands, or cellular (tissue/organ/whole body) functions that is governed by these substances is to develop exact models (equations) containing the relevant parameters of these substances or cellular (etc.) functions and then to fit them to properly prepared data that were obtained from biological measurements [3,4,5,6]. The first quantitative model of receptor function is the Hill equation, originally developed to describe the kinetics of a simple ligand binding [7], which relates the concentration of an agonist to the response evoked [4]. The Hill equation serves as the basis for all advanced quantitative (and semi-quantitative) receptor models [2,4,8], such as the widely used operational model of agonism [9] and the most recent SABRE (Signal Amplification, Binding affinity, and Receptor activation Efficacy) model [10]. Some models have been developed to address the special (rather than general) challenges, just like the receptorial responsiveness method (RRM), which combines the Hill equation with a simple relationship between simultaneous effects of two concentrations of one or two agonist(s), where one concentration is known and the other one is unknown (c_x_, for more detail, see below). Under certain circumstances, RRM can be used to estimate an acute increase in the level of an agonist (as c_x_) in the vicinity of its receptors in a biological system [11]. Technically, it is of great impact to possess reliable input data and ensure the most appropriate implementation of the fitting of the model of RRM.

RRM is a procedure that is based on a simple nonlinear regression while using a model with two variables (X, Y) and (at least) one parameter to be determined (c_x_) [11]. The fitting of this model requires two sets of concentration-effect (E/c) curves to be generated. E/c curves that are suitable for RRM are XY graphs where X is the (logarithm of the) concentration of a pharmacological agonist, while Y is a response of a biological system that is evoked by the given agonist concentration (indicated by the corresponding X value) alone (first set of curves) or together with a single extra concentration (c_x_) of the same or another agonist, which was administered to the system before the generation of the E/c curve (second set of curves). Quantifying this extra agonist concentration as a c_x_ value is the goal of RRM [11,12,13].

Although an E/c curve in the second set relates the resultant effect of the two concentrations (consuming the same response capacity of a biological system) solely to the concentration that is administered for the E/c curve (that is depicted on the *x*-axis), the model of RRM attributes this resultant effect to two concentrations, namely to two concentrations of the agonist that was used for the E/c curve. One of these concentrations is indicated by the *x*-axis, while the other one, c_x_, is a concentration of the agonist used for the E/c curve that is equieffective with the agonist added in a single extra dose. If the two agonists in question are the same, c_x_ is a real concentration, and, if not, c_x_ is a surrogate parameter of the single extra concentration of the other agonist [14,15]. The application of RRM might be useful when the concentration of this extra agonist is unknown and difficult to determine in any other manners [12,13]. As the single extra agonist concentration distorts (biases) the E/c curve in the second set as compared to the corresponding E/c curve in the first set (generated the same way except for the administration of the single extra agonist concentration), it will be referred to as “biasing” concentration.

This way, RRM can provide information regarding an acute increase in the concentration of an agonist in the microenvironment of its receptor. Importantly, the microenvironment of receptors is a tissue compartment that is difficult to access, especially in a moving organ [11,12,13]. Theoretically, RRM can be applied for each receptor; however, the A_1_ adenosine receptor is uniquely suitable for this method, due to its slow and incomplete desensitization in the presence of even a full agonist [16,17].

Regression analysis (especially its nonlinear form) is one of the most common ways to analyze E/c curves that might provide several useful pieces of information, e.g., about properties of receptors, receptor ligands and cell functions, which would be otherwise difficult (if possible) to gain [18,19,20]. The goal of regression (curve fitting) is to find the best-fit values for the parameters of the model that is used for the regression, and thereby to create the best-fit regression function (curve), which is closest to the data points (XY data obtained usually from repeated measurements). Minimizing the sum of the squares of the vertical distances of data points (replicate Y values related to the same X value) from the curve (the Y value of the curve that relates to the corresponding X value) is the earliest and most common technique to find best-fit values. Accordingly, this procedure is called the least-squares method. The use of this method is based on two major assumptions: (i) normality, i.e., the scatter of the data points (related to the same X value) around the curve (the Y value on the curve related to the corresponding X value) follows a Gaussian distribution; and, (ii) homoscedasticity, i.e., the extent of this scatter is the same for all values of X [18,19,20].

Ad (i): Assuming a Gaussian distribution for the scatter of data (and performing an ordinary regression) is useful for most cases. Nevertheless, assuming a Lorentzian distribution (and carrying out robust regression) makes the curve fitting more robust to outliers, although it hinders the calculation of data characterizing the reliability of the best-fit values and the curve (e.g., standard errors, confidence intervals, confidence, and prediction bands). In some cases, it is worth considering a Poisson distribution (and performing Poisson regression), but never for normalized Y data (used in the present study as well). While the Gaussian and Lorentzian distributions (being t distributions with infinity and 1 as degree(s) of freedom, respectively) allow for the use of the least-squares method, the Poisson distribution requires an alternative way for finding best-fit values (the so-called maximum likelihood-based parameter estimation) [18,19].

Ad (ii): To counteract heteroscedasticity, the standard equation of the least-squares method can be transformed (“weighted”) by a factor, a procedure that is called weighting. The most common factors are 1/Y^2^ (relative weighting) and 1/|Y| (Poisson weighting). The relative weighting and, to a lesser extent, the Poisson weighting reduce the influence of the higher Y values on the best-fit values and the regression curve. In addition, the Poisson weighting (performed with ordinary regression) can serve as an inferior alternative of the Poisson regression. A rarely used, although theoretically meaningful, choice is weighting by 1/SD^2^ (the inverse of the variance of Y values related to the same X value), which is expected to reduce the undue impact of Y replicates with bigger scatter [18,19].

The simplest and most common way of curve fitting is the individual regression, i.e., to find best-fit values for a single data set, e.g., a single E/c curve (or a set of E/c curves resulted from repeated measurements). An advanced way of curve fitting is when the model of regression defines a family of curves, i.e., some parameters (at least one) of the model to be fitted are (is) shared among several (at least two) data sets, called global regression. In the case of global regression, one sum of squares is computed for all of the Y replicates of all data sets [18,19].

The goal of the present study was to explore the influence that different curve fitting ways (individual vs. global fitting) and curve fitting settings (assuming different distributions and scatter patterns for the Y values) might exert on the outcome of RRM (Table 1). The experiments consisted of the construction of two E/c curves. For the first E/c curve, adenosine was administered to assess the naïve response of the atria to A_1_ adenosine receptor stimulation. Adenosine is particularly suitable for this purpose, because it is quickly eliminated without yielding confounding byproducts [21]. For the second E/c curve, one of three widespread, relatively stable, synthetic A_1_ adenosine receptor agonists (CPA, NECA, and CHA) was used, in the absence or presence of a “biasing” concentration of the same agonist. The accuracy and precision of RRM was investigated via assessing this known “biasing” concentration in a well-established isolated and paced guinea pig left atrium model. Accordingly, c_x_, estimate yielded by RRM, has been expected to directly provide the “biasing” concentration.

During this study, by measuring the left atrial contractile force, the negative inotropy that was elicited by the A_1_ adenosine receptor agonists was determined as an effect. In the ventricular myocardium of most mammalian species, adenosine receptor agonists fail to directly evoke a negative inotropic effect, i.e., without a previous increase of the cellular cAMP level [22]. In the mammalian atrium, however, stimulation of the A_1_ adenosine receptor can considerably reduce even the resting contractile force, exerting a significant direct negative inotropic effect [23,24]. As, for this study, paced left atria were used, the negative tropic effects that were mediated by the A_1_ adenosine receptor can manifest themselves only in a decrease of the resting contractile force. This feature has made our results more reliable and easier to interpret, as the direct negative inotropy is sensitive to any change in the frequency of contractions [25].

## 2. Results

### 2.1. Response to Adenosine

Adenosine concentration-dependently decreased the contractile force of atria in all groups (importantly, at this stage of the investigation, the groups were uniformly handled yet) (Figure 1). The corresponding empirical parameters of the adenosine E/c curves, provided by the fitted Hill model (Equation (1)), did not significantly differ from one another among the groups (data not shown). This result indicates the homogeneity of atria used for this study.

### 2.2. Responses to Synthetic A_1_ Adenosine Receptor Agonists

In the “Intact” groups, CPA, NECA, and CHA also decreased the atrial contractile force in a concentration-dependent manner (Figure 2a). The responses to the three synthetic agonists showed significant differences in the logEC_50_ (but not E_max_ and *n*) parameter (Table 2). In accordance with the fact that CPA, NECA, and CHA are worse substrates for adenosine-handling enzymes and carriers than adenosine [16,26], the EC_50_ values of the synthetic agonists were considerably smaller (by about two–three orders of magnitude) than EC_50_ of adenosine, with similar E_max_ values for all four agonists (cf. Figure 1 and Figure 2a).

In the “Biased” groups, in accordance with the expectations, a single large dose of CPA, NECA, or CHA (100 nM, 100 nM, or 300 nM, respectively, being greater by about one order of magnitude than the corresponding EC_50_ value) produced a substantial depression of the maximal response and a considerable dextral displacement of the E/c curve of the same agonist, when added before its construction (Figure 2b–d).

### 2.3. Curve Fitting to the Biased Responses Given to Synthetic A_1_ Adenosine Receptor Agonists

The outcome of regression has been characterized by the accuracy of logc_x_, the best-fit value (indicated by the nearness of its antilog (c_x_) to the corresponding “biasing” concentration) and by the precision of the curve fitting (as indicated by the 95% confidence interval (95% CI) of the best-fit value).

The c_x_ values proved to be acceptable estimates of the “biasing” concentrations in all cases when RRM was carried out without any weighting. In contrast, weighting by 1/SD^2^ and, especially, by 1/Y^2^, dramatically worsened the accuracy of estimates; moreover, in some cases, it hindered the curve fitting (Table 3). Narrow 95% CIs could be obtained when using ordinary regression (capable of yielding 95% confidence limits for the best-fit values) without weighting, if Equation (2) was individually fitted (i.e., separately to the E/c curves averaged within the groups; more precisely, separately to each set of replicate E/c curves forming one group). If not (namely, in case of global regression), the curve fitting provided wide 95% CIs (Table 3).

Surprisingly, in the present investigation, the global regression (simultaneous fitting of Equation (2) to the averaged E/c curves of the corresponding “Intact”-“Biased” E/c curves) did not improve the results of RRM. The global regression, although ensuring a quicker and more convenient way to obtain c_x_ values, yielded less accurate estimates than the conventional individual fitting (Table 3). Consistent with this, the global regression increased (rather than decreased) the uncertainty of the curve fitting, which produced wide 95% CIs (Table 3).

The robust regression (assuming Lorentzian distribution) moderately ameliorated the accuracy of the estimates in comparison with the ordinary regression (while assuming Gaussian distribution), except for the global and robust fitting in the Biased CHA group (Table 3). With the use of our curve fitting software, robust regression could not be combined with different weighting options.

In sum, weighting is of the greatest importance regarding the accuracy of RRM, whereas the assumption made on the type of distribution (whether Gaussian or Lorentzian) has the least impact among the investigated fitting options and ways. With the use of RRM, biological data of the present study were able to be analyzed the best if no weighting was implemented; furthermore, individual fitting was chosen rather than global one.

### 2.4. Curve Fitting to the Intact Responses Given to Synthetic A_1_ Adenosine Receptor Agonists

Curve fitting in the “Intact” groups served as an internal control for the curve fitting in the “Biased” groups. During individual regression, the “Intact” E/c curves were independently fitted from the “Biased” E/c curves. In contrast, the global regression provided c_x_ values for the corresponding “Intact” and “Biased” E/c curves at once. All the curve fittings used for internal control (that did not fail) yielded very small best-fit values (antilogs of which were near zero in most cases), as expected (Table 3).

Logarithm of zero is not defined, thus no definite best-fit value belongs to the fitting of Equation (2) to intact responses. This fact might be the cause of the large uncertainty of the curve fitting observed in the case of most “Intact” E/c curves (indicated by wide 95% CIs). Having narrow 95% CI for a best-fit value in an “Intact” group (e.g., individual and ordinary fitting with weighting by 1/Y^2^ in the Intact NECA group) appears to be a chance rather than a regular behavior (Table 3).

## 3. Discussion

The major finding of the present study is that the best estimates of RRM can be obtained via individual fitting without any weighting, almost irrespectively of the fact of whether ordinary (assuming Gaussian distribution) or robust (assuming Lorentzian distribution) regression is chosen. Thus, RRM is a relatively (by comparison to the possibilities it offers) easy-to-use procedure that requires neither a heavy-duty curve fitting software nor a high level of knowledge concerning regression analysis.

The model of RRM is a simplified mathematical description of the co-action of two agonists (evoking the same response of a biological system), if one of these agonists exerts its effect prior to the other one, and the former effect is disregarded (thereby causing a “bias”) [11,14]. RRM has been originally developed as a concentration determining method that is based on the fitting of its model (Equation (2)) to an E/c curve that was constructed in the presence of a “biasing” concentration that is to be determined. The model should contain the empirical parameters of the intact E/c relationship, where these data should be obtained from an E/c curve that is generated under the same conditions but in the absence of any “biasing” concentration. An important feature of RRM, opening up numerous opportunities, is that the agonist of the “biasing” concentration and the agonist used for the E/c curves can be different [11,12]. This way, RRM makes it possible to quantify an acute increase in the concentration of an agonist with a short half-life. For this purpose, E/c curves should be constructed with a similar but stable agonist in the same system. These E/c curves will be suitable for a quantitative evaluation by means of RRM, owing to the stability of the agonist used for them. In this case, RRM will result in a surrogate parameter of the “biasing” concentration, i.e., the equieffective concentration of the stable agonist. Thus, RRM can provide information regarding the change of agonist concentrations near the receptors, a compartment otherwise difficult to access, especially for agonists with short half-life.

Receptor desensitization is a general mechanism that affects every receptor, including the adenosine receptor family [16,17,27]. This phenomenon is of special significance regarding the A_1_ adenosine receptor, as any alteration in its responsiveness might modify vital protective and regenerative processes throughout the body [16,28]. Probably in strong association with this fact, the desensitization of the A_1_ adenosine receptor is slower than that of other receptors [16,17]. Owing to this, the A_1_ adenosine receptor (with its cellular environment) forms a biological system, in which RRM, which is a time-consuming (≈40 min.) procedure, can be successfully performed. Accordingly, although adenosine has a short half-life (seconds) in living tissues [26], RRM was first applied to estimate the interstitial accumulation of endogenous adenosine in the presence of a nucleoside transport inhibitor (dipyridamole or nitrobenzylthioinosine) in atria isolated from eu- and hyperthyroid guinea pigs. CPA equivalents of these surplus adenosine concentrations were, in fact, determined, as CPA was used to generate the necessary E/c curves [12,13,29,30,31].

Subsequently, the suitability of RRM has come to light in other fields of application as well. On one hand, the theoretical basis of RRM can be exploited when an agonist acutely accumulates in a tissue compartment that contains a receptor for this agonist. If the apparent function of the receptor (that can be detected by means of an E/c curve) does not follow the pattern predicted by the model of RRM, other mechanisms (e.g., receptor sensitization or desensitization, or change in activity of an enzyme, or transporter handling the given agonist) may be suspected. Such a phenomenon was revealed in an earlier study, where, via evaluating CPA E/c curves, it was found an enhancement of the efficiency of the A_1_ adenosine receptor signaling in the hyperthyroid guinea pig atrium under adenosine deaminase inhibition by pentostatin [32]. On the other hand, data provided by RRM can be applied to correct the E/c curves that are biased by an unknown (and thereby usually neglected) concentration of an agonist that has previously been accumulated in the system. The corrected adenosine E/c curves (constructed in the presence of nitrobenzylthioinosine) enabled us to qualitatively assess the receptor reserve for the negative inotropic effect of adenosine in eu- and hyperthyroid guinea pig atria in another earlier investigation [29,30]. It should be noted that the traditional methods to determine receptor reserve have not proved to be useful for adenosine due to the short half-life of adenosine [33], although receptor reserve values in the cardiac adenosinergic system seem to be of importance, even in terms of diagnostic aspects [34]. Finally, of course, data that were obtained with RRM can serve as input data for in silico investigations dealing with the RRM itself [14,15], and with the adenosinergic mechanisms in the atrial myocardium [35,36].

In all previous works, RRM was carried out with assumptions of Gaussian distribution for the scatter of data points around the curve, and of the same extent of this scatter along the curve. However, the long-established observation that these assumptions are usually true for non-transformed (at most normalized, in certain cases logarithmic) data obtained from biological systems was the only reason to do this [18,19,20]. Thus, one goal of the present study was to explore whether other curve fitting options based on other assumptions ameliorate the accuracy and/or precision (and thereby the usefulness and/or reliability) of RRM. In addition, the other goal of this investigation was to compare two ways of curve fitting, the individual and the global one. The global fitting is thought to be a powerful procedure that can reduce the uncertainty experienced when the corresponding E/c curves are individually fitted [18,19].

NECA, CPA, and CHA were decided as agonists to construct the E/c curves to be fitted in the present work. NECA stimulates both the A_1_ and A_2_ adenosine receptors, while CPA and CHA are highly selective for the A_1_ adenosine receptor [16,27], the predominant adenosine receptor type of the myocardium [16,24]. These synthetic compounds are less sensitive to adenosine-handling enzymes and carriers than adenosine, the endogenous agonist of the adenosine receptor family [16,26], so E/c curves of these synthetic agonists are more suitable for quantitative evaluation than those of adenosine.

The results of the present study show that the well-established observation, i.e., assumption of both normality and homoscedasticity is a useful (and first choice) approach when analyzing biological data, is valid for the assessment with RRM as well. Consequently, ordinary regression without any weighting is the appropriate decision, when performing RRM. Regarding only the accuracy of the fitting, robust regression can also be chosen (moreover, somewhat more accurate estimates may be obtained), but the lack of 95% CIs deprives the possibility of considering the precision (and thus reliability) of the curve fitting.

However, based on the present results, the conventional individual regression is a more accurate and precise (although less comfortable) way to carry out RRM than the global regression. This might be associated with the fact that the “intact” one of the two corresponding E/c curves has no definite best-fit value (logc_x_), because the expected estimate (c_x_) is zero, the logarithm of which is not defined. This problem does not occur when RRM is done with individual regression, because the “intact” E/c curve is fitted to the Hill equation (Equation (1)) that can yield reliable best-fit values (which ones will be then substituted into Equation (2) before the fitting of the “biased” E/c curve). Thus, the individual fitting is the appropriate choice for RRM.

## 4. Materials and Methods

### 4.1. Materials

As a bathing medium for the preparations, modified Krebs–Henseleit buffer (referred to as Krebs solution) was used with the following composition in g (MW in g/mol): NaCl: 69 (58.44); KCl: 3.5 (74.56); CaCl_2_∙2H_2_O: 3.7 (147.02); NaH_2_PO_4_: 1.56 (137.99); MgCl_2_∙6H_2_O: 2.7 (203.3); NaHCO_3_: 21 (84.01); glucose∙H_2_O: 22 (198.18); l-ascorbic acid: 0.1 (176.12); and, dissolved in 10 L of redistilled water.

As A_1_ adenosine receptor agonists, adenosine, *N*^6^-cyclopentyladenosine (CPA), 5′-(*N*-ethylcarboxamido)adenosine (NECA), and *N*^6^-cyclohexyladenosine (CHA), all being purchased from Sigma (St. Louis, MO, USA), were used. Adenosine was dissolved in 36 °C Krebs solution. CPA, NECA, and CHA were dissolved in ethanol:water (1:4) solution (*v*/*v*). All of the stock solutions were adjusted to a concentration of 10 mM. Stock solutions were diluted with Krebs solution.

### 4.2. Animals, Preparations and Protocols

The experiments were carried out by the approval of the Local Ethics Committee of Animal Research, University of Debrecen, Hungary (code: 25/2013/DEMÁB; 12 December 2013), in the spirit of the XXVIII of 1998 Act on the Protection and Welfare of Animals. Male Hartley guinea pigs, weighing 600–800 g, were guillotined and then the left atria were quickly removed and mounted at 10 mN resting tension in 10 mL vertical organ chambers (type: Experimetria TSZ-04; manufacturer: Experimetria Kft, Budapest, Hungary) filled with Krebs solution, aerated with 95% O_2_ and 5% CO_2_ (36 °C; pH = 7.4). Atria were electrically paced by platinum electrodes (3 Hz, 1 ms, twice the threshold voltage) by means of a programmable stimulator (type: Experimetria ST-02; manufacturer: Experimetria Kft, Budapest, Hungary) and power amplifier (type: Experimetria PST-02; manufacturer: Experimetria Kft, Budapest, Hungary). The amplitude of the isometric twitches (contractile force) was measured by means of a transducer (type: Experimetria SD-01; manufacturer: Experimetria Kft, Budapest, Hungary) and strain gauge (type: Experimetria SG-01D; manufacturer: Experimetria Kft, Budapest, Hungary), and recorded by a polygraph (type: Medicor R-61 6CH Recorder; manufacturer: Medicor Művek, Budapest, Hungary).

The atria were divided into six groups, according to the six experimental protocols that were carried out (Intact CPA group, Biased CPA group, Intact NECA group, Biased NECA group, Intact CHA group and Biased CHA group; *n* = 5–7).

In the organ chambers, all of the atria were first incubated for 40 min (in Krebs solution). Next, a cumulative E/c curve was constructed using adenosine (from 0.1 µM to 1 mM), followed by a washout period (Krebs solution for 15 min). Afterwards, in the “Intact” groups, a cumulative E/c curve was generated with CPA, NECA, or CHA (from 0.1 nM to 100 µM). Meanwhile, a single CPA, NECA, or CHA dose was administered to the atria in the Biased CPA, NECA, or CHA group to reach 100 nM, 100 nM, or 300 nM concentration (“biasing” concentration) in the bathing medium, respectively. Next, a cumulative E/c curve was constructed with the same agonist as was previously administered in a single dose, i.e., with CPA, NECA, or CHA (from 0.1 nM to 100 µM).

### 4.3. Empirical Characterization of the E/c Curves

All of the E/c curves were fitted to the Hill equation [4]:(1)$$E=E_{\max}\times \frac{c^{n}}{c^{n}+EC_{50}{}^{n}}\;$$
where: E: the effect that was defined as a percentage decrease in the initial contractile force of atria; c: the concentration of the agonist that was administered during the construction of the given E/c curve; E_max_: the maximal effect; EC_50_: the agonist concentration producing half-maximal effect (sometimes called as median-effective agonist concentration); and, *n*: the Hill coefficient (slope factor).

The individual and the averaged E/c curve data were fitted to the Hill equation for the statistical analysis and to illustrate the E/c curves, respectively.

### 4.4. Assessment of the “Biasing” Concentration

The CPA, NECA, and CHA E/c curves (averaged within the groups) were fitted to the model of RRM:(2)$$E^{\prime }=100-\frac{100\times \left(100-E_{\max}\times \frac{{\left(c_{x}+c\right)}^{n}}{{\left(c_{x}+c\right)}^{n}+EC_{50}{}^{n}}\right)}{100-E_{\max}\times \frac{c_{x}{}^{n}}{c_{x}{}^{n}+EC_{50}{}^{n}}}\;$$
where: E’: the biased effect (effect distorted by a “systematic error”, c_x_, see below), which was calculated from the raw data in a conventional manner (i.e., regardless of whether a “biasing” concentration was present); E_max_, EC_50_, *n*: empirical parameters of the intact E/c relationship according to the Hill model (Equation (1)); c: the concentration of the agonist administered during the construction of the E/c curve; and, c_x_: the “biasing” concentration (the estimate provided by RRM).

The Equation (2) was fitted two ways: individually and globally. During the individual regression, Equation (2) was fitted to the averaged E/c curve, generated with a synthetic agonist, of each group in a manner that the appropriate empirical parameters were previously obtained by fitting the Hill equation (Equation (1)) to the averaged E/c curve of an “Intact” group that was constructed with the same synthetic agonist. This means that, for the fitting of each averaged CPA, NECA, and CHA E/c curve (in either an “Intact” or a “Biased” group), Equation (2) had to be individualized by substituting the appropriate empirical parameters in it. In turn, upon global regression, Equation (2) was simultaneously fitted to the averaged E/c curves of the corresponding “Intact” and “Biased” groups, sharing their empirical parameters (E_max_, EC_50_, and *n*).

As described in the previous paragraph, during the individual regression, Equation (2) was also fitted to the E/c curves in the “Intact” groups. This served as an internal control (because the correct estimate, zero, was known in advance); furthermore, it ensured a better comparability with the results of the global regression (since the global fitting inherently contains such an internal control, yielding c_x_ values for both the ”Intact” and “Biased” E/c curves at once).

### 4.5. Fitting Settings

In addition to the two fitting ways that were described in the previous subsection (individual vs. global regression), the fitting of Equation (2) was performed while using the following setting options: ordinary vs. robust regression; furthermore, a lack of weighting vs. weighting by 1/Y^2^ vs. weighting by 1/SD^2^. The different ways and setting options were combined with one another in all possible manners (Table 1).

Replicate Y (effect) values were always considered as individual points, as recommended (except for the case of weighting by 1/SD^2^, where the mean of Y replicates could only be considered). For every other setting, the default option was used [19].

### 4.6. Data Analysis

Each atrium had to meet three criteria for inclusion in the evaluation: (i) the resting contractile force had to reach 1 mN before the adenosine E/c curve; (ii) the mechanical activity of the paced atrium had to be regular; and, (iii) the response to 10 µM adenosine was required to be within the mean ± 2SD range. The mean and SD were computed while using atria obeying the first two criteria.

More than two data sets were compared with one-way ANOVA, followed by Tukey post-testing, after the verification of the Gaussian distribution of data with Shapiro–Wilk test.

Concentrations (c, EC_50_, and c_x_) in the equations used for curve fitting were expressed as common logarithms (logc, logEC_50_, and logc_x_), as recommended [18,19]. Statistical analysis and curve fitting were performed with GraphPad Prism 8.2.1 for Windows (GraphPad Software Inc., La Jolla, CA, USA), while other calculations were made by means of Microsoft Excel 2016 (Microsoft Co., Redmond, WA, USA).

## Figures and Tables

**Figure 1 ijms-20-06264-f001:**
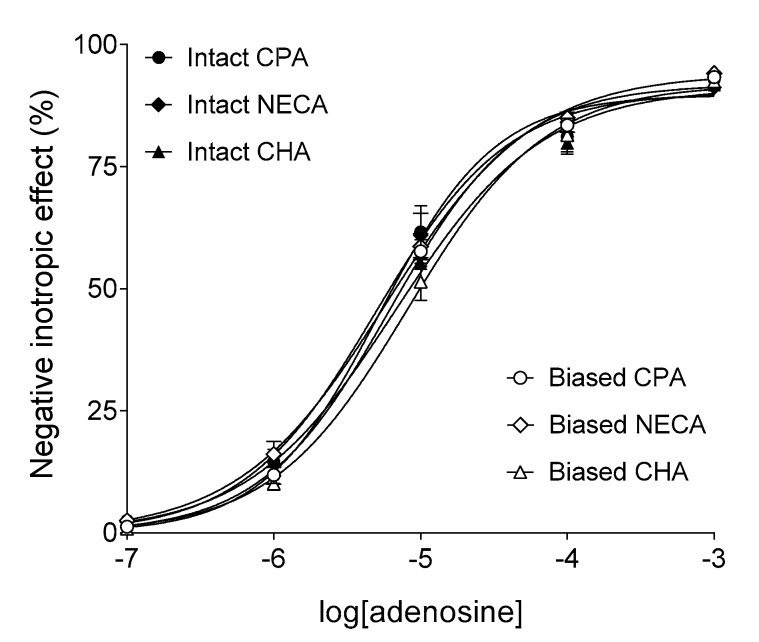
The direct negative inotropic response of isolated guinea pig left atria to adenosine, in the six groups. The *x*-axis denotes the common logarithm of the molar concentration of adenosine (administered during the construction of the E/c curve). The *y*-axis shows the effect (percentage decrease in the initial contractile force). The terms “Intact” (filled symbols) and “Biased” (open symbols) in the group names refer to a subsequent (and not the current) condition. The responses to adenosine were averaged within the groups and indicated by the symbols (±SEM). The curves illustrate the Equation (1) (Hill model) fitted to the averaged responses. CPA: *N*^6^-cyclopentyladenosine; NECA: 5′-(*N*-ethylcarboxamido)adenosine; CHA: *N^6^*-cyclohexyladenosine.

**Figure 2 ijms-20-06264-f002:**
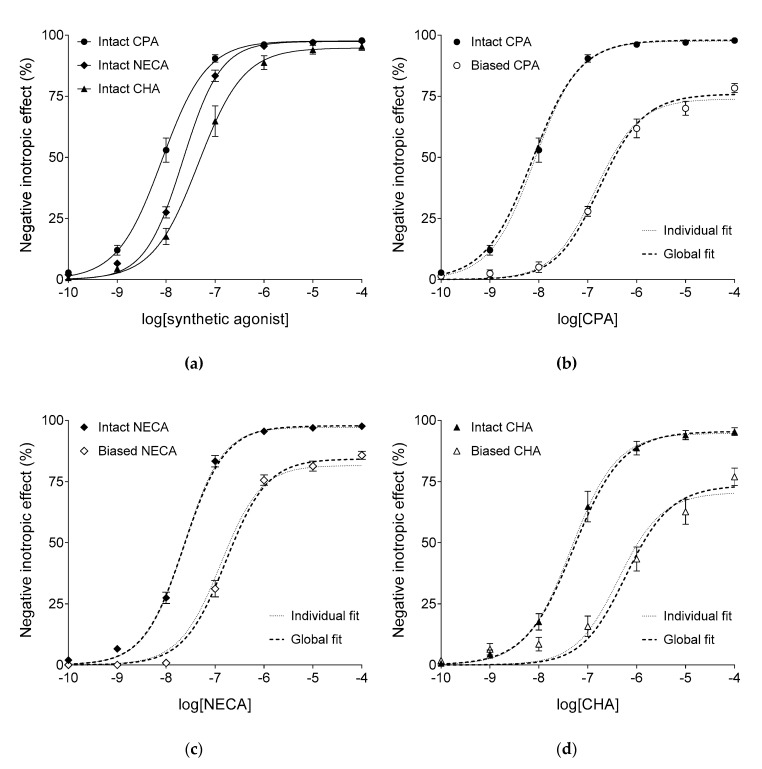
The direct negative inotropic response of isolated guinea pig left atria to three synthetic A_1_ adenosine receptor agonists, in the six groups. The *x*-axis denotes the common logarithm of the molar concentration of the given agonist (administered during the construction of the E/c curve). The *y*-axis shows the effect (percentage decrease in the initial contractile force). Atria in the “Intact” groups (filled symbols) underwent a conventional concentration-effect (E/c) curve construction, while atria in the “Biased” groups (open symbols) received a surplus dose (of the agonist indicated below the *x*-axis) before the generation of the E/c curve. All “Intact” groups are presented in the panel (**a**), while E/c curves of CPA, NECA and CHA are separately shown in panels (**b**–**d**), respectively. The symbols indicate the responses to the agonists averaged within the groups (±SEM). The continuous lines show the fitted Equation (1) (Hill model). The thin dotted lines denote the individually fitted Equation (2) (the model of RRM), while the thick dotted lines illustrate the globally fitted Equation (2). Settings for Equation (2) for both ways of fitting were robust regression with no weighting (providing, in general, the most accurate estimates). CPA: *N*^6^-cyclopentyladenosine; NECA: 5′-(*N*-ethylcarboxamido)adenosine; CHA: *N^6^*-cyclohexyladenosine.

**Table 1 ijms-20-06264-t001:** Combination of two fitting ways with two setting options addressing the data distribution and with three further setting options addressing the data homo- or heteroscedasticity (reflecting the properties of our curve fitting software, i.e., the lack of weighting during robust fitting [19]).

	Ordinary Fit	Robust Fit
**No weighting**	Individual fit	Individual fit
	Global fit	Global fit
**Weighting by 1/Y^2^**	Individual fit	
	Global fit	
**Weighting by 1/SD^2^**	Individual fit	
	Global fit	

**Table 2 ijms-20-06264-t002:** Empirical data (mean ± SEM) of the concentration-effect curves in the “Intact” groups (seen in the Figure 2a).

	Intact CPA (*n* = 6)	Intact NECA (*n* = 7)	Intact CHA (*n* = 6)
E_max_ (%)	97.4 ± 0.8	97.5 ± 0.5 ns	94.8 ± 1.7 ns, ns
logEC_50_	−8.1 ± 0.09	−7.66 ± 0.05 **	−7.33 ± 0.11 ###, **+**
*n*	1.02 ± 0.05	1.14 ± 0.07 ns	0.98 ± 0.03 ns, ns

All significant differences are indicated (*: CPA vs. NECA; #: CPA vs. CHA; **+**: NECA vs. CHA). The number of marks refers to the level of statistical significance (one mark: *p* < 0.05; two marks: *p* < 0.01; three marks: *p* < 0.001). CPA: *N^6^*-cyclopentyladenosine; NECA: 5′-(*N*-ethylcarboxamido)adenosine; CHA: *N^6^*-cyclohexyladenosine; ns: non-significant.

**Table 3 ijms-20-06264-t003:** Best-fit values (logc_x_) with their 95% confidence intervals (95% CI) and antilog values (c_x_) in all the six groups.

**Individual, Ordinary, ø**	**Biased CPA**	**Intact CPA**	**Biased NECA**	**Intact NECA**	**Biased CHA**	**Intact CHA**
logc_x_	−6.88	−19,107	−6.87	−8.92	−6.45	−52,574
95% CI	−6.93 to −6.83	very wide	−6.92 to −6.83	? to −8.32	−6.55 to −6.36	very wide
c_x_ (nM)	131.4	0	133.6	1.2	352.7	0
**Individual, Robust, ø**	**Biased CPA**	**Intact CPA**	**Biased NECA**	**Intact NECA**	**Biased CHA**	**Intact CHA**
logc_x_	−6.9	−19,107	−6.88	−8.64	−6.47	−5257
c_x_ (nM)	125.9	0	131.4	2.3	335.8	0
**Global, Ordinary, ø**	**Biased CPA**	**Intact CPA**	**Biased NECA**	**Intact NECA**	**Biased CHA**	**Intact CHA**
logc_x_	−6.84	−7201	−6.77	−35,907	−6.39	−9,892,707,770
95% CI	very wide	very wide	very wide	very wide	very wide	very wide
c_x_ (nM)	145.9	0	170	0	403.7	0
**Global, Robust, ø**	**Biased CPA**	**Intact CPA**	**Biased NECA**	**Intact NECA**	**Biased CHA**	**Intact CHA**
logc_x_	−6.85	−7201	−6.82	−35,907	−6.36	−9,892,707,770
c_x_ (nM)	142.6	0	153.3	0	439	0
**Individual, Ordinary, 1/Y^2^**	**Biased CPA**	**Intact CPA**	**Biased NECA**	**Intact NECA**	**Biased CHA**	**Intact CHA**
logc_x_	−8.21	−19107	−6.36	−8.51	−317,820,174,071	−52,574
95% CI	? to −8.163	very wide	? to −6.34	−8.89 to −8.2	very wide	very wide
c_x_ (nM)	6.2	0	438.5	3.1	0	0
**Global, Ordinary, 1/Y^2^**	**Biased CPA**	**Intact CPA**	**Biased NECA**	**Intact NECA**	**Biased CHA**	**Intact CHA**
logc_x_	−366.5	−7201	−6.146	−35,907	-	-
95% CI	very wide	very wide	very wide	very wide		
c_x_ (nM)	0	0	714.8	0		
**Individual, Ordinary, 1/SD^2^**	**Biased CPA**	**Intact CPA**	**Biased NECA**	**Intact NECA**	**Biased CHA**	**Intact CHA**
logc_x_	−6.9	−42,781	-	−8.66	−6.5	−56,039
95% CI	−6.99 to −6.82	very wide		? to −7.96	−6.77 to −6.26	very wide
c_x_ (nM)	125.7	0		2.2	318.6	0
**Global, Ordinary, 1/SD^2^**	**Biased CPA**	**Intact CPA**	**Biased NECA**	**Intact NECA**	**Biased CHA**	**Intact CHA**
logc_x_	−6.87	−1353	-	-	−8.26	−22.51
95% CI	very wide	very wide			very wide	very wide
c_x_ (nM)	136.5	0			554.1	3.11 × 10^−14^

ø: non-weighted; Y: Y value (effect); SD: standard deviation; CPA: *N^6^*-cyclopentyladenosine; NECA: 5′-(*N*-ethylcarboxamido)adenosine; CHA: *N^6^*-cyclohexyladenosine. In some cases, no results could be obtained.

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
