# Peer review of "Accuracy and Precision of the Receptorial Responsiveness Method (RRM) in the Quantification of A1 Adenosine Receptor Agonists"

_ijms, 2019, doi:10.3390/ijms20246264_

Round 1
Reviewer 1 Report
Receptor desensitization is a general mechanism affecting every receptor including the adenosine receptor family. This phenomenon is of special significance regarding the A1 adenosine receptor, as any alteration in its responsiveness may modify vital protective and regenerative processes throughout the body. The goal of the present study was to explore the influence that different curve fitting settings (assuming different distributions and scatter patterns for the Y values) and ways (individual vs. global fitting) might exert on the outcome of RRM. Herein, RRM was used to assess known concentrations of three widespread synthetic A1 adenosine receptor agonists (CPA, NECA and CHA) in a well-established isolated and paced guinea pig left atrium model.
Results of the present study show that the best estimates of RRM can be obtained via individual fitting without any weighting, almost irrespectively of the fact whether ordinary (assuming Gaussian distribution) or robust (assuming Lorentzian distribution) regression is chosen. Thus, RRM is a procedure that requires neither a heavy-duty curve fitting software nor high level of knowledge concerning regression analysis.
Comments: The ABSTRACT shall be improved by adding a sentence on the biological significance of this study, what was the reason to conduct this study. INTRODUCTION shall be extended with the state-of-the art in the field.
Author Response
Response to Reviewer 1
Thank You for reviewing our manuscript. The Abstract has been completed and the Introduction has been extended, as recommended. Overall, the manuscript was reworked to improve its clarity. We hope that you will find our revised manuscript suitable for publication in the IJMS.
Reviewer 2 Report
In the introduction the concept has to be explained in figure representation. This would help the readers to understand the concept in more easily.
Author Response
Response to Reviewer 2
Thank You for reviewing our work. The Introduction has been extended; a new Table has also been inserted to improve the clarity of the concept of the manuscript. Overall, the manuscript underwent an extensive revision. We hope that you will find the present version of our manuscript suitable for publication in the IJMS.
Round 2
Reviewer 1 Report
The authors have adequately addressed my questions